# TRIM6: An Upregulated Biomarker with Prognostic Significance and Immune Correlations in Gliomas

**DOI:** 10.3390/biom13091298

**Published:** 2023-08-24

**Authors:** Jianrong Guo, Shoucheng Feng, Hong Liu, Zhuopeng Chen, Chao Ding, Yukai Jin, Xiaojiang Chen, Yudong Ling, Yi Zeng, Hao Long, Haibo Qiu

**Affiliations:** 1State Key Laboratory of Oncology in South China, Department of Gastric Surgery, Collaborative Innovation Center for Cancer Medicine, Sun Yat-Sen University Cancer Center, Guangzhou 510060, China; guojr1@sysucc.org.cn (J.G.); dingchao@sysucc.org.cn (C.D.); jinyk@sysucc.org.cn (Y.J.); chenxiaoj1@sysucc.org.cn (X.C.); lingyd@sysucc.org.cn (Y.L.); cengyi@sysucc.org.cn (Y.Z.); 2State Key Laboratory of Oncology in South China, Department of Thoracic Surgery, Collaborative Innovation Center for Cancer Medicine, Sun Yat-Sen University Cancer Center, Guangzhou 510060, China; fengsc@sysucc.org.cn; 3State Key Laboratory of Oncology in South China, Department of Neurosurgery, Collaborative Innovation Center for Cancer Medicine, Sun Yat-Sen University Cancer Center, Guangzhou 510060, China; liuhong@sysucc.org.cn (H.L.); chenzp@sysucc.org.cn (Z.C.)

**Keywords:** TRIM6, gliomas, prognostic biomarker, cytokine-cytokine receptor interaction, immune infiltrates

## Abstract

This study investigates the expression and prognostic value of TRIM6 in gliomas, the most prevalent primary brain and spinal cord tumors. Our results show that TRIM6 is predominantly overexpressed in glioma tissues and is associated with reduced overall survival, disease-specific survival, and progression-free interval. Furthermore, TRIM6 expression is correlated with WHO grade and primary treatment outcomes. Functional analysis indicates that interactions between cytokines and their receptors play a critical role in the prognosis of glioma patients. A protein-protein interaction network reveals 10 hub genes closely linked to cytokine-cytokine receptor interaction. In vitro experiments demonstrate that silencing TRIM6 impairs the proliferation, invasion, and migration of glioma cells, while overexpressing TRIM6 enhances these abilities. Additionally, TRIM6 expression is positively associated with the abundance of innate immune cells and negatively associated with the abundance of adaptive immune cells. In summary, TRIM6 is significantly upregulated in gliomas and linked to poor prognosis, making it a potential diagnostic and prognostic biomarker. TRIM6 plays a crucial role in promoting cell viability, clonogenic potential, migration, and invasion in glioma cells. It may regulate glioma progression by modulating cytokine-cytokine receptor interaction, leading to an inflammatory response and an imbalance in immunomodulation, thereby representing a potential therapeutic target.

## 1. Introduction

Gliomas are the most common type of primary brain and spinal cord tumor, accounting for 81% of malignancies in the central nervous system (CNS) [1,2]. Histologically, they resemble normal glial cells and are named accordingly [1]. Gliomas typically originate from glial or precursor cells and develop into astrocytoma, oligodendroglioma, ependymoma, or oligoastrocytoma [1,3,4]. In recent years, advances in cancer genetics and molecular characterization have greatly expanded our understanding of glioma biology. According to the World Health Organization (WHO) classification, gliomas are divided into four grades, with grades 1 and 2 indicating low-grade gliomas and grades 3 and 4 indicating high-grade gliomas (HGG) [3]. Generally, the higher the grade, the poorer the prognosis [5]. Pilocytic astrocytoma (grade I) has the highest 5-year relative survival rate of approximately 95% [5], while glioblastoma, the most common glioma histology (accounting for approximately 45% of all gliomas), has a 5-year relative survival rate of approximately 5% [6,7]. It has recently been discovered that glioma patients with isocitrate dehydrogenase (IDH) mutations have a relatively favorable prognosis [8,9]. Gliomas are also 40–50% more common in adult males than in females [10]. Adults over the age of 65 have the highest incidence of higher-grade and more aggressive gliomas, while lower-grade and less aggressive forms are more common in younger adults, particularly those between the ages of 20 and 40 [10]. Despite the discovery of various cancer drugs in recent decades, few have been approved by the Food and Drug Administration (FDA) for the treatment of gliomas. One reason for this lack of progress is the blood-brain barrier, which consists of endothelial cells, capillaries, and basement membranes. This unique structure in the CNS prevents most antitumor drugs from entering the brain, posing challenges for the development of antiglioma drugs [11].

Tripartite motif-containing proteins (TRIM), of which there are over 70, play crucial roles in immune responses, cancer growth, and chemoresistance [12,13,14]. Tripartite motif-containing protein 6 (TRIM6) is a member of the TRIM family of proteins. The TRIM6 gene is located on chromosome 11p15 and is part of a cluster of TRIM genes that also includes the TRIM5, TRIM21, TRIM22, TRIM34 genes, and a TRIM pseudogene [15]. Like other members of the TRIM family, TRIM6 has a tripartite motif and exhibits E3-ubiquitin ligase activity [16]. Previous research has identified roles for TRIM6 in viral infection and inflammatory responses. According to Rajsbaum et al., TRIM6 can activate IKK and enhance the induction of type I interferon (IFN-I)-stimulated genes (ISGs), facilitating the regulation of viral infection [16]. Van Tol S et al. showed that depletion of TRIM6 in human cells results in increased West Nile Virus (WNV) replication and alters the expression and function of other components of the IFN-I pathway through VAMP8 [17]. Shuier Zheng et al., reported that TRIM6 promotes colorectal cancer cell proliferation and response to thiostrepton via TIS21/FoxM1 [18]. However, the potential involvement of TRIM6 in cancer development has received little attention.

In this study, we demonstrate that TRIM6 expression is significantly upregulated in glioma samples and investigate the association between TRIM6 expression and clinical characteristics of glioma patients. To better understand the role of the TRIM6 gene in gliomas, we analyzed the correlation between TRIM6 expression and patient prognosis using data from The Cancer Genome Atlas (TCGA). Additionally, we conducted Gene Ontology (GO), Kyoto Encyclopedia of Genes and Genomes (KEGG), and Gene Set Enrichment Analysis (GSEA) analyses to identify potential pathways and processes. Furthermore, we validated the expression of TRIM6 and 9 key genes closely related to TRIM6 in gliomas using RT-qPCR experiments. Moreover, we experimentally confirmed in vitro that knockdown of TRIM6 can inhibit the proliferation, invasion, and migration abilities of glioma cells, while overexpression of TRIM6 can enhance these abilities. Finally, we investigated the relationship between TRIM6 and tumor-infiltrating lymphocytes (TILs). Our findings suggest that TRIM6 may serve as a biomarker for predicting the prognosis and immune infiltration of individuals with gliomas.

## 2. Materials and Methods

### 2.1. RNA Expression and Data Mining

Patient Datasets: We downloaded messenger RNA (mRNA) expression data, including 689 glioma samples and 1157 non-tumor samples, from the TCGA database (https://cancergenome.nih.gov, accessed on 12 January 2023.) and the GTEx database (https://commonfund.nih.gov/gtex, accessed on 12 January 2023.). The data were extracted in TPM format, and clinical information was obtained from the TCGA database. Our data filtering strategy involved removing normal samples and samples without clinical information. The data processing method used was log2(value + 1). Our study was conducted in accordance with the publication guidelines provided by TCGA. To further validate our findings, we also downloaded gene expression profiles from the Gene Expression Omnibus (GEO) database (https://www.ncbi.nlm.nih.gov/geo/, accessed on 12 January 2023.), including GSE109569 (comprising 3 glioma samples and 3 paired adjacent non-tumor samples) and GSE76070 (comprising 3 glioma samples and 3 paired adjacent non-tumor samples). We used the Human Protein Atlas (HPA) database (http://www.proteinatlas.org/, accessed on 12 January 2023.) to verify the expression of TRIM6 in glioma at the protein level.

### 2.2. Survival Analysis

Based on the median value of TRIM6 expression levels, TCGA glioma data were divided into high and low TRIM6 groups. We used the Kaplan-Meier method and a two-sided log-rank test to analyze differences in overall survival (OS) between the high-risk and low-risk groups. The Stats [4.2.1], survival [3.3.1], survminer, and car packages in R were used to estimate the correlation between TRIM6 expression and the survival rate of various clinical features in glioma patients, and the hazard ratio (HR) and log-rank *p*-value of the 95% confidence interval were calculated.

### 2.3. Construction and Prediction of the Nomogram

In this study, we used Cox regression analysis to select all independent clinicopathological prognostic factors and generated a contingency table to analyze the 1, 3, and 5 year OS probabilities of glioma patients using the rms package in R. The survival package is used for proportional risk hypothesis testing and Cox regression analysis. Variable screening strategy: samples in a single factor meet the set *p*-value (0.1) threshold and enter the multi-factor Cox to build the model. Calibration and discrimination are the most common methods for evaluating model performance. Receiver operating characteristic (ROC) analysis was used to assess the predictive accuracy of the combined model’s line chart compared to the line charts of other clinicopathological prognostic variables.

### 2.4. GO, KEGG and GSEA Analysis

We used the DESeq2 [19] package to analyze differentially expressed genes (DEGs) between the high and low TRIM6 expression groups in glioma patients. DEGs were identified using an unpaired *t*-test, with a threshold value of adjusted *p* < 0.05 and |logFC| > 2, calculated using the Benjamini-Hochberg method. GO analysis revealed that these genes were represented in various functional categories, including biological processes, molecular functions, and cellular components. KEGG enrichment and pathway analyses of DEGs were conducted using the Database for Annotation, Visualization, and Integrated Discovery (DAVID) online tools (https://david.ncifcrf.gov). The KEGG pathway database is a resource for understanding the high-level functions and utilities of biological systems, including various biochemical pathways. In this study, we used the clusterProfiler package [20] for GO and KEGG analyses.

### 2.5. Protein Interaction PPI Network Construction and Hub Genes Analysis

The STRING Database (https://string-db.org/, accessed on 14 January 2023.) is a search engine for known protein-protein interactions. After obtaining data from the STRING database, you can use Cytoscape, an open-source network visualization and analysis application. Its primary goal is to provide a basic functional layout and query network, and to construct a protein-protein interaction (PPI) network by combining basic data into a visual network. The Cytoscape plugin cytoHubba (version 0.1) can identify hub genes in the PPI network.

### 2.6. Tissue Samples Collection

Four glioma samples and corresponding non-tumor tissue samples were obtained from Sun Yat-sen University Cancer Center (SYSUCC). Written informed consent was obtained from all patients enrolled in the study. All experiments using clinical samples were carried out under protocols approved by the institutional review board at Sun Yat-sen University Cancer Center. Approval Code: B2022-536-01.

### 2.7. Cell Culture and Reagents

The U251 and U373 cell line was maintained in the State Key Laboratory of Oncology in South China of SYSUCC (Guangzhou, China). Cells were grown in DMEM medium (Thermo Fisher Scientific, Waltham, MA, USA) supplemented with 10% fetal calf serum (FBS), 100 μg/mL Penicillin, and 100 μg/mL Streptomycin at 37 °C in a humidified incubator containing 5% carbon dioxide.

### 2.8. Cell Transfection and RNA Knockdown

To study TRIM6’s function, we performed cell transfection and RNA knockdown experiments. Lentiviral construct pEZ-Lv241 containing the full length TRIM6 (GeneCopoeia, Germantown, MD, USA) was packaged into 293T cells using the ViraPower Mix (Invitrogen, Carlsbad, CA, USA) according to the manufacturer’s instructions. U373 cells were stably transfected with TRIM6-expressing lentivirus or lentiviral vector plus 10 mg/mL polybrene (Beyotime Biotechnology, Haimen, China). To establish TRIM6 knockdown cells, short hairpin RNAs (shRNA) in lentivirus against TRIM6 (GeneCopoeia, Germantown, MD, USA) were stably transduced into U251 cells. Confirmed siRNA targeting sequences and the reference gene were listed in Appendix A.

### 2.9. RNA Preparation and Quantitative RT-qPCR

Total cellular RNA was extracted using TRI Reagent (Invitrogen, Carlsbad, CA, USA) follow the manufacturer’s instructions. cDNA was synthesized using Hifair^®^ III 1st Strand cDNA Synthesis SuperMix (Yeasen Biotechnology, Shanghai, China). Then, RT-qPCR was performed using Hieff^®^ qPCR SYBR^®^ Green Master Mix (Yeasen Biotechnology, Shanghai, China) on the LightCycler^®^ 480 Real-Time PCR System to determine the mRNA expression of targeted genes. The amplification of glyceraldehyde-3-phosphate dehydrogenase (GAPDH) gene was used as an internal control. The primer sequences used for analysis are shown in Appendix A.

### 2.10. Antibodies and Western Blot Analysis

Western blot analysis was performed according to the standard protocol with antibodies against TRIM6 (FNab08991) obtained from FineTest; β-actin (#4967) obtained from Cell Signaling Technology.

### 2.11. Cell Proliferation and Foci Formation Assay

Cell Counting Kit-8 (CCK-8) revealed the presence of cellular proliferation (Dojindo, Shanghai, China). The cells were grown in the 96well plate (3 × 10^3^ cells/well) in the 37 °C CO**_2_** incubator for 0 h, 24 h, 48 h, 72 h and 96 h before 10 μL of the CCK-8 mixture was added to each well. After incubating the cells at 37 °C for 2 h, the absorbance was measured at 450 nm using a microplate reader. In foci formation assay, 1 × 10^3^ cells were seeded in six-well plate. After 14 days culture, cell colonies were counted by crystalviolet staining.

### 2.12. Wound Healing Assay and Transwell Assay

The capacity for cell migration was detected using wound healing assay. Cells were seeded into a six well plate (5 × 10^5^ per well) and cultured to 90% confluence before 200 μL pipette tips were used to make scratches. The six well plates were then cultured for 48 h at 37 °C and 5% CO**_2_** in a cell culture incubator. The migratory distance of the cells was captured using the FSX100 BioImage System (Media Cybernetics, Rockville, MD, USA) and analyzed using Imagepro plus at 0 and 48 h after wound scratching. The percentage of wound closure was determined as (wound closure area/initial area) 100 percent. (* *p* < 0.05)

The Transwell assay was performed using 24-well Transwell inserts (8 μm pore size). The U251 and U373 cells were cultured in DMEM supplemented with FBS and penicillin-streptomycin. The Transwell inserts were coated with or without Matrigel, and 1 × 10^5^ cells were seeded onto the upper chamber. The lower chamber contained DMEM supplemented with FBS as a chemoattractant. The plates were incubated at 37 °C and 5% CO**_2_** for 48 h. After incubation, the Transwell inserts were removed, and migrated cells on the lower surface were fixed, stained, and imaged. Cell quantification was performed using imageJ software. Statistical analysis was conducted using Student’s *t*-test.

### 2.13. Statistical Analysis

To assess the degree of TRIM6 gene expression in patients with Gliomas, box plots and scatter plots were used. The median method of gene expression was determined to be the TRIM6 expression cutoff value. The relationship between TRIM6 expression and the clinical features of Gliomas was examined using the Wilcoxon signed-rank test and logistic regression. The log-rank test was used to look at the *p*-value. The chi-square test was used for categorical data, whereas the t test was used for numerical variables. To find pertinent predictive factors, both univariate and multivariate Cox analyses were used. In all analyses, *, **, and *** indicate *p* < 0.05, *p* < 0.01 and *p* < 0.001, respectively.

## 3. Results

### 3.1. Results

#### 3.1.1. TRIM6 Is Highly Expressed in Gliomas

To investigate the expression level of TRIM6 in tumor and normal tissue, we analyzed TRIM6 mRNA expression in various cancers and normal tissues using data from the TCGA and GTEx databases. The GTEx database served as a control unrelated to TCGA. We accessed TPM-formatted RNAseq data from TCGA and GTEx, which were imported into UCSC Xena (https://xenabrowser.net/datapages/, accessed on 12 January 2023.) and processed using the universally accepted Toil procedure [21]. The results demonstrated significantly higher TRIM6 expression in Glioma tissue compared to normal tissues (Figure 1A). Furthermore, subgroup analysis indicated that both GBM and LGG patients exhibited significantly elevated TRIM6 expression compared to normal tissue (Figure 1B,C).

#### 3.1.2. Validation Using Independent External Databases and Clinical Specimens

To further validate the expression level of TRIM6 in Gliomas, we utilized two additional independent external GEO datasets (validation cohort), namely GSE109569 and GSE76070, to analyze TRIM6 transcription levels in cancer tissues and adjacent tissues of Gliomas. The analysis revealed a significant increase in TRIM6 transcription levels in Gliomas compared to normal adjacent tissues, as evident from both GSE109569 (*p* < 0.05) and GSE76070 datasets (*p* < 0.001) (Figure 1D,E). To corroborate the protein expression level of TRIM6 in Gliomas, we examined the HPA database, which confirmed higher TRIM6 expression levels in glioma tumor tissues compared to healthy cerebral cortex tissues. Moreover, the TRIM6 expression levels exhibited an increasing trend with WHO grade (Figure 1G). Additionally, to strengthen our findings, we conducted qPCR analysis on tumor and adjacent non-tumor tissue specimens obtained from four glioma patients at our institution. The results further supported our observations, showing a significant upregulation of TRIM6 mRNA expression levels in the glioma tumor tissues when compared to the adjacent non-tumor tissues (Figure 1F).

#### 3.1.3. Association of the Expression of TRIM6 and Clinicopathologic Factors

We analyzed the mRNA expression levels of TRIM6 in various clinical categories using the TCGA database to explore the association between TRIM6 expression and clinical features in glioma patients. Table 1 provides a summary of the TCGA dataset, consisting of 698 tumor samples and 5 normal samples. To perform survival analysis based on TRIM6 expression levels, we stratified these glioma patients into two groups: the TRIM6 high-expression group and the TRIM6 low-expression group, using the median TRIM6 expression level as the cutoff. Our results revealed significant correlations between high TRIM6 expression and several clinical characteristics of these patients. Specifically, TRIM6 high expression was significantly associated with Gender (*p* < 0.05), Age (*p* < 0.001), Primary therapy outcome, WHO grade (G2 vs. G3, *p* < 0.01; G2 vs. G4, *p* < 0.001; G3 vs. G4, *p* < 0.001), IDH status (*p* < 0.001), histological type, and 1p/19q codeletion (*p* < 0.001) (Figure 2A–G). The data suggest that TRIM6 is notably upregulated in Gliomas, particularly in male patients (Figure 2A) and patients over 60 years old (Figure 2B). Additionally, patients with high TRIM6 expression were more likely to have progression disease (PD) at the primary therapy outcome (Figure 2C). The expression level of TRIM6 increased with the WHO grade, indicating a positive correlation between higher grade gliomas and elevated TRIM6 expression (Figure 2D). Furthermore, the expression level of TRIM6 was higher in wild type Glioma patients compared to IDH-Mut Gliomas patients (Figure 2E). We also observed variations in TRIM6 expression among different histological types of Gliomas, with the highest expression in Glioblastoma, followed by astrocytoma and oligoastrocytoma, and the lowest expression in oligodendroglioma (Figure 2F). Additionally, Glioma patients with 1P/19Q deletion mutations exhibited lower expression levels of TRIM6 compared to those without 1P/19Q deletion mutations (Figure 2G). We also found that the expression levels of TRIM6 showed no statistically significant difference in OS among patients with WHO grade G4, histologically classified as Oligodendroglioma and Glioblastoma, as well as those who had primary therapy outcome categorized as complete response (CR) or partial response (PR) (Appendix A).

#### 3.1.4. TCGA Gliomas Dataset Analysis Reveals That TRIM6 Expression Is Associated with Reduced Patient Survival

The OS study demonstrated that Glioma patients with high TRIM6 expression had a significantly poorer prognosis than those with low TRIM6 expression (*p* < 0.001) (Figure 3A–C). The disease-specific survival (DSS) and progression-free interval (PFI) analyses yielded consistent results with the OS analysis, further indicating that TRIM6 expression is associated with worse survival in Glioma patients. Subgroup analysis revealed a strong correlation between high TRIM6 expression and poor prognosis in various clinical categories. Specifically, the following cases showed a significant association with high TRIM6 expression and adverse outcomes in Gliomas: patients under 60 years old (HR = 3.33, *p* < 0.001), patients over 60 years old (HR = 1.79, *p* = 0.05), male patients (HR = 4.24, *p* < 0.001), female patients (HR = 3.67, *p* < 0.001), WHO grade G2 (HR = 2.1, *p* = 0.033), WHO grade G3 (HR = 2.51, *p* < 0.001), patients with PD (progression disease; HR = 2.11, *p* = 0.001), patients with SD (stable disease; HR = 2.12, *p* = 0.033), patients with histological type Astrocytoma (HR = 3.79, *p* < 0.001), and patients with histological type Oligoastrocytoma (HR = 2.49, *p* = 0.033) (Figure 3D–M).

#### 3.1.5. Diagnostic and Prognosistic Value of TRIM6 Expression in Glioma Patients

According to the TCGA database, TRIM6 expression can be considered a discriminatory factor based on the ROC analysis, with an area under the curve (AUC) of 0.821, indicating its potential as a diagnostic biomarker for Gliomas relative to normal tissue (Figure 4A). The AUC for TRIM6 expression in Glioblastoma (GBM) and Lower Grade Glioma (LGG) is 0.961 and 0.791, respectively (Figure 4B–C). Furthermore, ROC curve and nomogram analyses were conducted on TRIM6 gene expression data from the TCGA database to evaluate its prognostic value. The AUCs for 1-year, 3-year, and 5-year OS were 0.776, 0.817, and 0.746, respectively (Figure 4D). The AUCs for 1-year, 3-year, and 5-year DSS were 0.783, 0.817, and 0.754, respectively (Figure 4E). The AUCs for 1-year, 3-year, and 5-year PFI were 0.759, 0.734, and 0.774, respectively (Figure 4F). These findings indicate that high TRIM6 expression is a significant poor prognostic indicator for Glioma patients. Cox regression analyses were performed to identify independent predictors of OS in Gliomas. In the univariate model, WHO grade, IDH status, 1p/19q codeletion, primary therapy outcome, age, histological type, and TRIM6 expression were all significantly associated with OS in Gliomas (Table 2, all *p* < 0.05). In the multivariate analyses, TRIM6, as well as WHO grade, IDH status, primary therapy outcome, gender, and age, remained significantly associated with OS in Gliomas (HR = 1.591, 95% CI = 1.027–2.466, *p* = 0.038). The DSS analysis yielded similar results (Appendix A, HR = 1.716, 95% CI = 1.080–2.726, *p* = 0.022). However, the multivariate regression analysis did not show a significant association between TRIM6 and PFS in Glioma patients (Appendix A, HR = 1.090, 95% CI = 0.766–1.551, *p* = 0.633). These findings suggest that TRIM6 expression is an independent predictive factor for Glioma patients’ OS and DSS. A nomogram was constructed using the expression level of TRIM6 and clinical factors confirmed by multivariate studies to predict the survival probability of patients at 1, 3, and 5 years (Figure 4G). The calibration plot demonstrated good agreement between the prediction and the observation, as the bias-corrected line closely approximated the ideal 45-degree line (Figure 4H).

#### 3.1.6. GO, KEGG and GSEA Enrichment Analysis of TRIM6-Related DEGs

To investigate the potential role of TRIM6 in the development of Gliomas, we compared the gene expression profiles of TRIM6 high-expression (*n* = 351) and low-expression (*n* = 350) groups using RNAseq. A total of 1123 upregulated genes and 26 downregulated genes were detected in the TRIM6 high-expression group compared to the TRIM6 low-expression group (Figure 5A). Functional enrichment analysis using DAVID revealed that these DEGs were enriched in various biological processes, cellular components, and molecular functions, including regionalization, pattern specification process, collagen-containing extracellular matrix, cytokine activity, receptor ligand activity, and signaling receptor ligand activity (Figure 5B,C). KEGG enrichment analysis indicated that the DEGs were significantly enriched in pathways such as Cytokine-cytokine receptor interaction, IL-17 signaling pathway, and Viral protein interaction with cytokine and cytokine receptor (Figure 5C). To gain further insight into the biological pathways involved in Glioma pathogenesis based on TRIM6 expression level, GSEA was performed. The enrichment plots revealed that gene signatures related to Cytokine-cytokine receptor interaction, Hematopoietic cell lineage, Systemic lupus erythematosus, Allograft rejection, Leishmania infection, ECM receptor interaction, Complement-and-coagulation cascades, Autoimmune thyroid disease, and Jak-Stat signaling pathway were activated in patients with high TRIM6 expression (Figure 6A). Notably, the Calcium signaling pathway was downregulated among the top 10 enriched pathways (Figure 6B,C). Pathview analysis of the Cytokine-cytokine receptor interaction pathway further demonstrated the substantial activation of this pathway by TRIM6 (Figure 6D). These findings suggest that TRIM6 may act as a tumor promoter in Gliomas by activating multiple signaling pathways, with the Cytokine-cytokine receptor interaction pathway being one of the most significantly affected pathways.

#### 3.1.7. The Establishment and Analysis of PPI Network Using TRIM6-Related DEGs

Data on protein interactions are represented by complex network diagrams in the STRING database, where nodes correspond to proteins and edges represent interactions between proteins. We constructed the PPI network diagram of DEGs using the STRING database (Figure 7A). Utilizing Cytoscape’s plug-in cytoHubba with the Degree algorithm, we identified the top 10 hub genes, namely COL1A1, COL1A2, CXCL8, CXCL9, CXCL10, CXCL11, CCL11, CXCR3, MMP9, and TIMP1, which exhibited the strongest associations with other nodes in the PPI network (Figure 7B). Gene co-expression correlation analysis demonstrated a strong positive correlation between these hub genes and TRIM6, as well as among themselves (Figure 7C). Subsequently, we collected tumor specimens from four patients with high-grade gliomas, along with their corresponding adjacent non-tumor tissues, at our institution. Using qPCR analysis, we evaluated the mRNA expression levels of the ten hub genes and TRIM6 in the tumor tissues relative to the adjacent non-tumor tissues. The results showed a significant upregulation of COL1A1, COL1A2, CXCL8, CXCL9, CXCL10, CXCL11, CCL11, CXCR3, MMP9, and TIMP1 mRNA expression levels in the glioma tumor tissues compared to the adjacent non-tumor tissues (Figure 7D).

#### 3.1.8. OS Analysis of Hubgenes in TCGA Glioma Patients

To validate the prognostic value of these hub genes, we performed OS analysis using Kaplan-Meier curves in TCGA Glioma patients. The results revealed that high expression of nine hub genes, namely COL1A1, COL1A2, CXCL8, CXCL9, CXCL10, CXCL11, CXCR3, MMP9, and TIMP1, was associated with worse survival outcomes among Glioma patients (Figure 8A–I). These findings contribute to a deeper understanding of Glioma pathogenesis and aid in the identification of potential therapeutic targets and prognostic biomarkers.

#### 3.1.9. TRIM6 Modulation Influences Glioma Cell Behavior In Vitro

We initially assessed the expression levels of TRIM6 in two glioma cell lines, U251 and U373, using qPCR analysis (Figure 9A). Remarkably, U251 cells exhibited higher TRIM6 mRNA transcript levels compared to U373 cells. To further investigate the functional role of TRIM6, we employed two different shRNAs to knockdown TRIM6 in U251 cells, while U373 cells were transfected with a TRIM6 overexpression plasmid. The transcriptional levels of TRIM6 were validated using qPCR, while TRIM6 protein expression levels were assessed by Western blotting. Notably, both shTRIM6-1 and shTRIM6-2 successfully led to a significant reduction in TRIM6 mRNA transcript levels (Figure 9B) and a corresponding decrease in protein expression levels (Figure 9D) compared to the shNC control group in U251 cells. Conversely, in U373 cells, the overexpression of TRIM6 following plasmid transfection resulted in a substantial upregulation of TRIM6 mRNA transcript levels (Figure 9C) and a significant increase in protein expression levels (Figure 9E) compared to the Vector control group. Subsequent CCK8 assays demonstrated a significant decrease in cell viability in both shTRIM6-1 and shTRIM6-2 groups of U251 cells after 96 h of TRIM6 knockdown, compared to the shCtrl control group (Figure 9F, *p* < 0.001). In contrast, the overexpression of TRIM6 in U373 cells led to a significant increase in cell viability compared to the Vector control group (Figure 9G, *p* < 0.001). Furthermore, we conducted colony formation assays. The results revealed a remarkable reduction in colony formation in the shTRIM6-1 and shTRIM6-2 groups of U251 cells compared to the shNC group (Figure 9H). Conversely, the overexpression of TRIM6 in U373 cells significantly enhanced colony formation compared to the Vector control group (Figure 9I). The cell wound healing assay and Transwell assay were performed to assess the impact of TRIM6 on the invasive and migratory abilities of glioma cells. The results are as follows: In the U251 cell line, both the shTRIM6-1 and shTRIM6-2 groups exhibited significantly wider scratch gaps compared to the control group (shNC) after 48 h, indicating decreased cell migration (Figure 10A,B). In the Transwell assay, the shTRIM6-1 and shTRIM6-2 groups showed a significant decrease in the invasion and migration abilities compared to the control group (shNC), indicating reduced cell invasion and migration (Figure 10E–G, *p* < 0.001). In the U373 cell line, the overexpression of TRIM6 resulted in a significant reduction in scratch gap width compared to the Vector control group after 48 h, indicating enhanced cell migration (Figure 10C,D). The overexpression of TRIM6 also resulted in a significant increase in the invasion and migration abilities compared to the Vector control group, indicating enhanced cell invasion and migration (Figure 10H–J, *p* < 0.001).

#### 3.1.10. TRIM6 Is Closely Related to Cell Cycle Regulatory Genes in Gliomas

We further analyzed the correlation between TRIM6 and the cell cycle regulation genes in TCGA-Gliomas, and we found that the expressions of the cell cycle regulatory genes CCNA2, CCNB1, CCNB2, CCNE1, CHEK1, BUB1B, ESPL1, PTTG1, PCNA, PKMYT1, CDC45, PLK1, MCM2, MCM4, MCM6, E2F1, CDC6, CDC20, CDC25A, and CDC25C were positively correlated with TRIM6 (r > 0.3, *p* < 0.001) (Figure 11A–J). These data verify that TRIM6 is closely associated with the regulation of the cell cycle in Gliomas.

#### 3.1.11. Correlation between TRIM6 Expression and Immune Characteristics

Research has shown that the TRIM family plays critical roles in immune responses, carcinogenesis, and chemoresistance [12,13,14]. In this study, we observed abnormal high expression of TRIM6 in Gliomas, which led us to speculate that TRIM6 might be involved in regulating the tumor immune response. To explore this hypothesis, we analyzed the correlation between the expression of TRIM6 and immune cell enrichment using the Spearman correlation and ssGSEA (single-sample Gene Set Enrichment Analysis) method. The results revealed a positive correlation between TRIM6 expression and the abundance of innate immune cells, including Macrophages, Neutrophils, Eosinophils, aDC, iDC, Cytotoxic cells, CD56dim Natural Killer cells (NK cells), Th2 cells, NK cells, and B cells (*p* < 0.05; Figure 12A). Conversely, TRIM6 expression showed a negative correlation with the abundance of adaptive immunocytes, such as pDC, Tgd, NK CD56bright cells, CD8 T cells, Tcm, and TFH (*p* < 0.05; Figure 12A). Further analysis demonstrated that Glioma patients with high TRIM6 expression had significantly higher infiltration levels of CD8+ T cells, dendritic cells (DCs), eosinophils, interdigitating cells, macrophages, mast cells, aDC, Cytotoxic cells, Eosinophils, iDC, Macrophages, Neutrophils, NK CD56dim cells, NK cells, Tgd, Th17 cells, and Th2 cells compared to patients with low TRIM6 expression (Figure 12B). In contrast, the infiltration of CD8 T cells, NK CD56bright cells, pDC, Tcm, TFH, and Tgd was significantly lower in Glioma patients with high TRIM6 expression compared to those with low TRIM6 expression (Figure 12B). However, there was no significant difference in the infiltration levels of B cells, Mast cells, Tem, and Treg between patients with high and low TRIM6 expression (Figure 12B). These findings indicate that the TRIM6 gene may play an important role in tumor immunity, particularly in modulating the infiltration of various immune cell types in Gliomas.

## 4. Discussion

TRIM proteins are well known to involve in immune responses and carcinogenesis [22]. TRIM6 has a tripartite motif and possesses E3-ubiquitin ligase activity like other TRIM family members [16]. The functions of TRIM6 in viral infection and inflammatory responses have been identified in earlier investigations. However, the role of TRIM6 in the development of cancer has rarely been reported. The mechanism of Gliomas’ tumorigenesis remains unknown. The relationship between viral infection and Glioma is one of the most important research fields [23]. Anna E Coghill et al., reported evidence of an inverse association between exposure to Epstein-Barr virus (EBV) and glioma and cytomegalovirus exposure may be related to a higher likelihood of the nonglioblastoma subtype [24]. Zehao Cai et al., reported the degree of Cytomegalovirus (CMV), Human Papillomavirus (HPV) and Human Herpesvirus 6 (HHV-6) infection infection have a significant impact on the prognosis of glioma patients in a meta-analysis [23].

According to the 2016 WHO classification, glioma is first classified according to histological features, and then more subtypes are classified according to molecular characteristics. There are a variety of indicators that are widely used in clinical practice (such as GFAP, EMA, MGMT, P53, NeuN, Oligo2, EGFR, VEGF, IDH1, Ki-67, 1p/19q), and these indicators are highly correlated with the prognosis of the patients [25,26,27]. Our results demonstrated that the expression of TRIM6 was significantly up-regulated in Gliomas samples compared to normal tissues, suggesting that TRIM6 may be a suitable target for the development of diagnostic techniques for patients with Gliomas and may be exploited in therapeutic settings. In addition, a high TRIM6 expression in Gliomas is associated with poor OS, DSS, and PFI. Moreover, the expression of TRIM6 in Gliomas was related with advanced clinicopathological features, according to our findings (WHO grade, Histological type, age sex and Primary therapy outcome). Regarding the relationship between TRIM6 expression and survival outcomes in glioblastoma and WHO Grade 4 patients, our analysis did not reveal a significant correlation. Although this lack of statistical significance may initially seem surprising, it is essential to consider the context of these findings. Grade 4 gliomas, including Glioblastoma, are known to exhibit exceptionally poor prognostic outcomes compared to lower grade tumors. Therefore, it is possible that the already pronounced adverse prognosis associated with Grade 4 gliomas could overshadow any potential association between TRIM6 expression levels and survival. While high TRIM6 expression has been linked to worse survival outcomes in other tumor types examined in our study, its impact on prognosis appears distinct within the context of Glioblastoma. This observation suggests that additional factors or molecular alterations might play more dominant roles in determining survival outcomes for these aggressive tumor types. Further investigation is warranted to comprehensively understand the underlying mechanisms involved in glioblastoma progression and how they intersect with TRIM6 function. By elucidating such complexities, we can gain valuable insights into potential therapeutic strategies targeting both TRIM6-related pathways as well as other key determinants of glioblastoma prognosis. 

Expression of TRIM6 was associated with clinical and pathological indicators of a poor prognosis, as determined by univariate analysis utilizing logistic regression. We employed univariate and multivariate analysis to determine the impact of TRIM6 expression on Glioma patients. These findings strongly suggest that TRIM6 may be exploited as an oncogene and prognostic biomarker. Our findings are in agreement with the conclusions derived from the study of Shuier Zheng et al. apparently supporting the tumor-promoting effect of TRIM6 [18]. However, further study is necessary to reach a definite conclusion.

To explain the underlying molecular mechanism by which TRIM6 influences the prognosis of Gliomas, we compared the gene expression profiles between TRIM6 high- and low-expression groups using the TCGA-GBMLGG database. DEGs were detected. In the category ‘MF’, the DEGs were clearly enriched in the categories of ‘cytokine activity’, ‘receptor ligand activity’, and ‘signaling receptor ligand activity’, which indicated that TRIM6 may play a molecular function by regulating cytokines and receptors. Cytokines and their receptors play an important role in immunomodulation, inflammatory response, tumor metastasis and other physiological and pathological processes [28,29], and also play an important role in the directional migration of immune cells [30,31,32,33]. KEGG enrichment analysis found that the DEGs were mostly enriched in ‘Cytokine-cytokine receptor interaction’, which supports the above conclusion. Moreover, ‘IL-17 signaling pathway’ and ‘Viral protein interaction with cytokine and cytokine receptor’ were also enriched. GSEA result indicated that Cytokine-cytokine receptor interaction signaling pathway was one of the most activated pathways. We defined 10 genes, including COL1A1, COL1A2, CXCL8, CXCL9, CXCL10, CXCL11, CXCR3, MMP9, and TIMP1, from the TRIM6 gene network as our hub genes. To our surprise, the high expression of these 10 hub genes was associated with poor prognosis of Gliomas. Moreover, the expression of these 10 hub genes was positively correlated with TRIM6. Notably, most of them are important genes in the Cytokine-cytokine receptor interaction pathway.

Our findings from the CCK8 assay demonstrated that the knockdown of TRIM6 in U251 glioma cells resulted in a significant decrease in cell viability compared to the control group. This suggests that TRIM6 may play a role in promoting cell survival and proliferation in U251 cells. The colony formation assay further supported the inhibitory role of TRIM6 knockdown in U251 cells. The decreased number of colonies formed by the shTRIM6-1 and shTRIM6-2 groups compared to the shNC group suggests that TRIM6 knockdown suppressed the clonogenic potential of U251 cells. This finding is consistent with the observed decrease in cell viability and indicates that TRIM6 may contribute to the proliferative capacity of U251 cells. In the scratch assay, we observed that the knockdown of TRIM6 in U251 cells resulted in a wider scratch gap compared to the control group, indicating impaired cell migration. The Transwell assay provided additional evidence for the involvement of TRIM6 in glioma cell invasion and migration. In U251 cells, the knockdown of TRIM6 significantly reduced the number of invading and migrating cells compared to the shNC group. Conversely, in U373 cells, the overexpression of TRIM6 led to a significant decrease in invasion and migration abilities compared to the Vector control group. These results are consistent with previous studies that have implicated TRIM6 in cancer cell growth and survival [18].

Based on our study findings, we observed a positive correlation between TRIM6 expression and the expression of several cell cycle regulation genes, including CCNA2, CCNB1, CCNE1, etc. While these findings suggest a potential association between TRIM6 and cell cycle progression in gliomas, further investigations are required to determine if TRIM6 directly regulates the cell cycle.

Tumor microenvironment (TME) is closely related to the development of cancer [34]. TME contains immune cells, extracellular matrix, mesenchymal cells, and inflammatory mediators that affect tumor growth, metastasis, and clinical survival [26]. Despite the fact that an effective immune response can have antitumor effects, cancer cells have evolved a variety of mechanisms, including a dysfunction in antigen presentation and a recruitment of immune suppressors to evade the attack of immune cells [35,36,37]. A number of studies have found that immune infiltration can affect the prognosis of patients [38,39,40]. We explored the correlation between the expression of TRIM6 and the level of immune infiltration of Gliomas. The present study found that the TRIM6 expression was positively correlated with the abundance of innate immune cells (eg, Macrophages, Neutrophils, Eosinophils, aDC, iDC, Cytotoxic cells, CD56dim NK cells, Th2 cells, NK cells and B cells), and negatively correlated with the abundance of adaptive immunocytes (eg, pDC, Tgd, NK CD56bright cells, CD8 T cells, Tcm and TFH). These findings suggest that the TRIM6 gene may indeed play a crucial role in tumor immunity by influencing the infiltration of different immune cell types in gliomas. By exploring the relationship between TRIM6 expression and tumor immune response, our study provides valuable insights into this area of research. However, it is important to note that further studies are needed to fully understand and validate these observations. Additional investigations can help confirm the functional significance of TRIM6 in modulating immune cell infiltration and elucidate its underlying mechanisms.

## 5. Conclusions

Taking it together, the current study found that TRIM6 expression is significantly upregulated and related to poor prognosis in Glioma patients. For the diagnosis and prognosis of gliomas, TRIM6 has certain reference values. TRIM6 plays a significant role in promoting cell viability, clonogenic potential, migration, and invasion in glioma cells. TRIM6 may regulate the progression of Gliomas by regulating the Cytokine-cytokine receptor interaction, thus enhances the inflammatory response, affecting immunomodulation imbalance. These findings support the potential of TRIM6 as a therapeutic target for inhibiting glioma progression and warrant further investigation into its underlying mechanisms and functional interactions in glioma pathogenesis. This study has some limitations. Firstly, despite concluding that TRIM6 expression is strongly associated with Cytokine-cytokine receptor interaction, immune infiltration and Gliomas’ prognosis, we lack direct evidence that TRIM6 influences prognosis through Cytokine-cytokine receptor interaction or immune infiltration. Future research should address these issues.

## Figures and Tables

**Figure 1 biomolecules-13-01298-f001:**
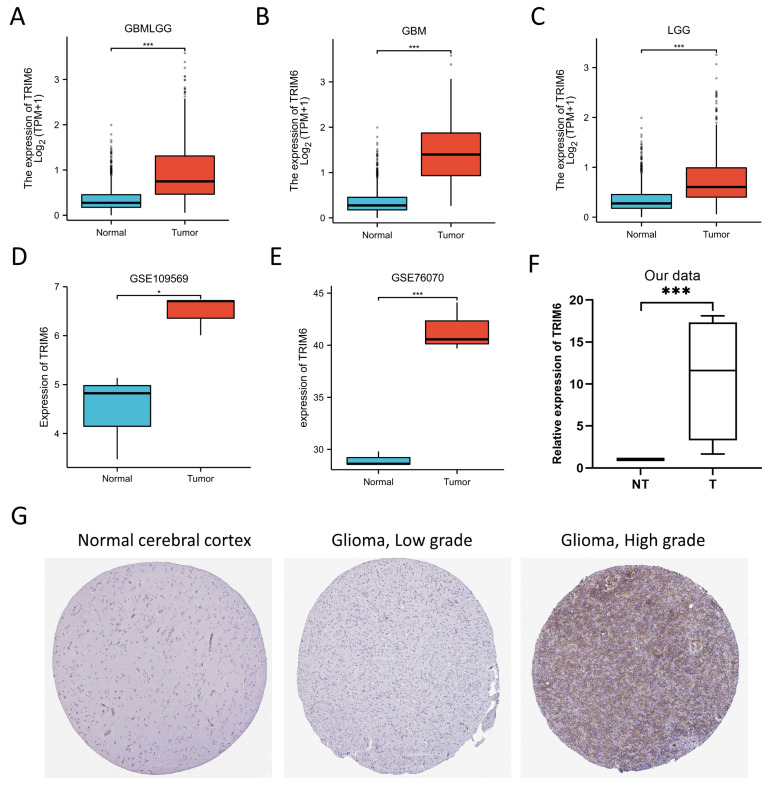
Expression analysis and validation of TRIM6. (**A**–**C**) Differences in TRIM6 expression between normal and Gliomas, GBM and LGG tissues in TCGA database. (**D**–**E**) showed that the transcription level of TRIM6 in Gliomas compared with in normal adjacent tissues from GSE109569 and GSE76070 datasets. (**F**,**G**) Validation of the expression level of TRIM6 between normal cerebral cortex and Gliomas tissues using four cases of glioma specimens collected in our center and the Human Protein Atlas database (immunohistochemistry). Data are presented as mean ± deviation (SD). Statistical analysis was performed using *t*-test, and *p* < 0.05 was considered statistically significant (* *p* < 0.05, *** *p* < 0.001). box-and-whisker plot displays the median (middle line within the box), quartiles (box boundaries), and potential outliers (individual data points).

**Figure 2 biomolecules-13-01298-f002:**
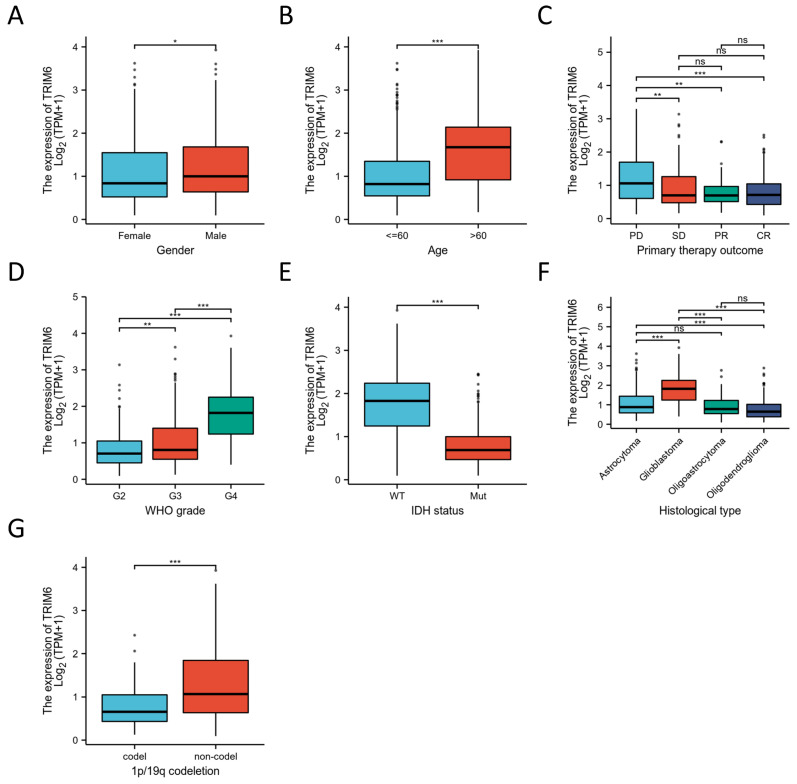
The correlation of the expression of TRIM6 and clinical features in Gliomas. Gender (**A**), Age (**B**), primary therapy outcome (**C**), WHO grade (**D**), IDH status (**E**), Histological type (**F**) and 1p/19q codeletion (**G**). Data are presented as mean ± SD. Statistical analysis was performed using *t*-test, and *p* < 0.05 was considered statistically significant (* *p* < 0.05, ** *p* < 0.01, *** *p* < 0.001). ns, non-significance. The interpretation of the box plot is consistent with Figure 1.

**Figure 3 biomolecules-13-01298-f003:**
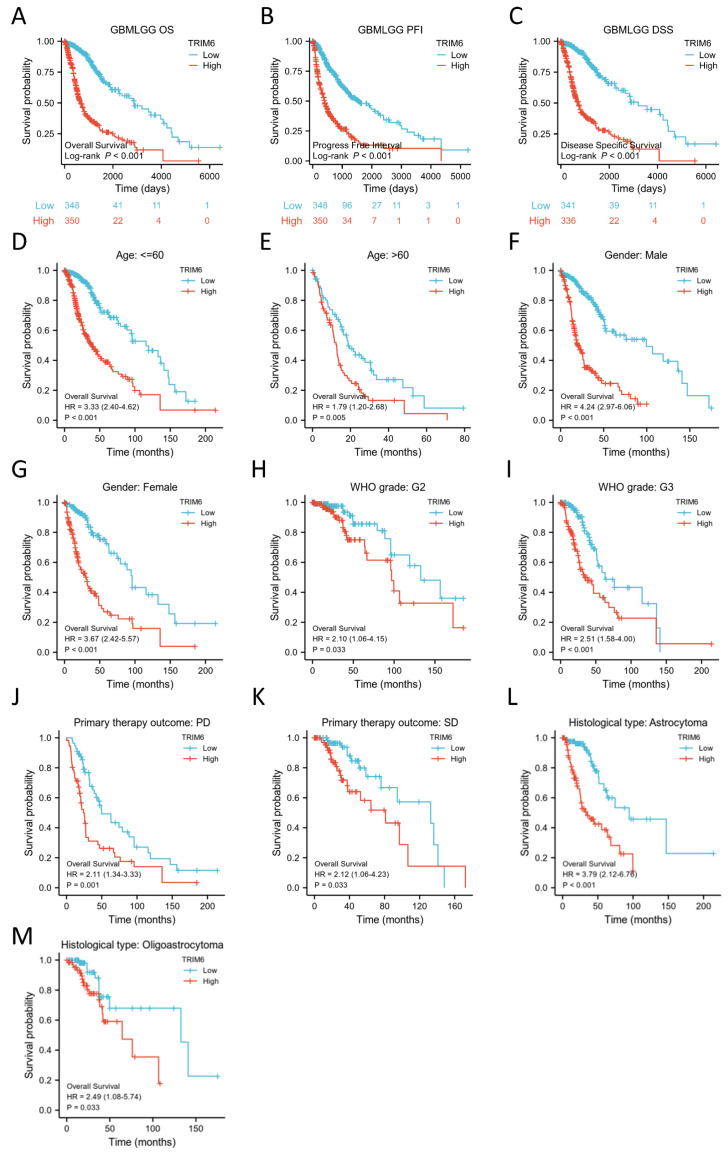
Kaplan–Meier survival curve analysis of the prognostic significance of a high and a low expression of TRIM6 in Gliomas using The Cancer Genome Atlas (TCGA) databases. (**A**–**C**) Kaplan–Meier estimates of the overall survival, disease specific survival and progress free interval probability of TCGA patients in all Gliomas patients. Subgroup analysis for age under 60 years (**D**), greater than 60 years (**E**), Male (**F**), Female (**G**), WHO grade G2 (**H**), WHO grade G3 (**I**), progression disease/stable disease (PD/SD) (**J**,**K**), Astrocytoma/Oligoastrocytoma (**L**,**M**).

**Figure 4 biomolecules-13-01298-f004:**
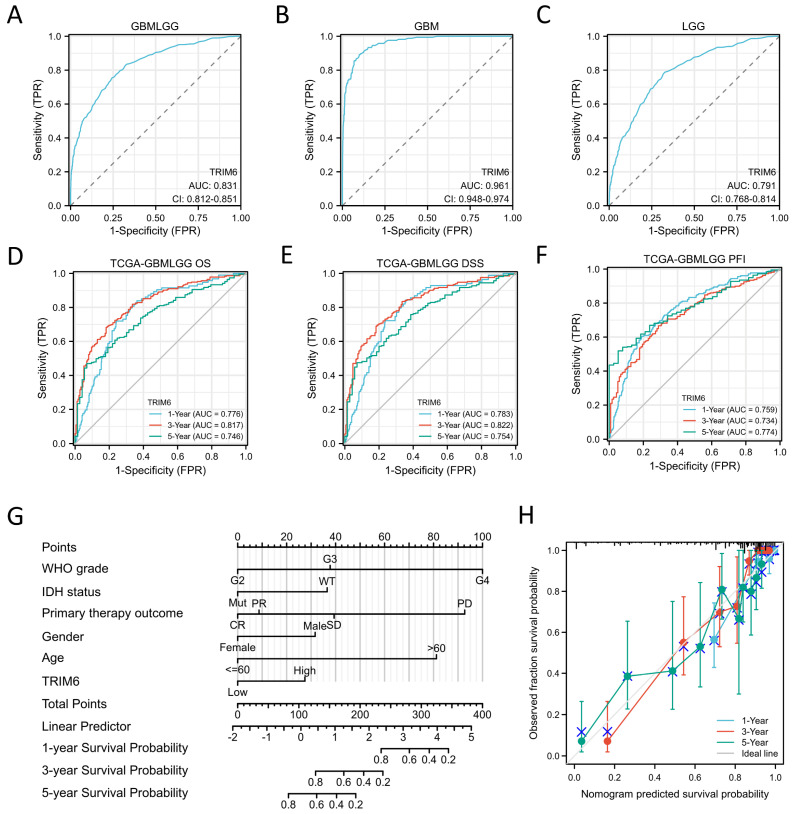
Diagnostic and prognosistic value of TRIM6 expression in Gliomas. (**A**–**C**) Validation of diagnostic value of TRIM6 upregulation for Gliomas, GBM and LGG using ROC curve. (**D**–**F**) Time depended ROC curve analysis of overall survival (OS), disease specific survival (DSS) and progress free interval (PFI) for TRIM6 expression in Gliomas. (**G**) Nomogram survival prediction chart for predicting the 1, 3, and 5year OS using the risk scores and clinical features in Gliomas. (**H**) Calibration curve predicting OS. Blue line in (**A**–**C**) represents the performance of TRIM6, gray line in (**A**–**F**) represents the performance of a classifier that uses a random guessing strategy.

**Figure 5 biomolecules-13-01298-f005:**
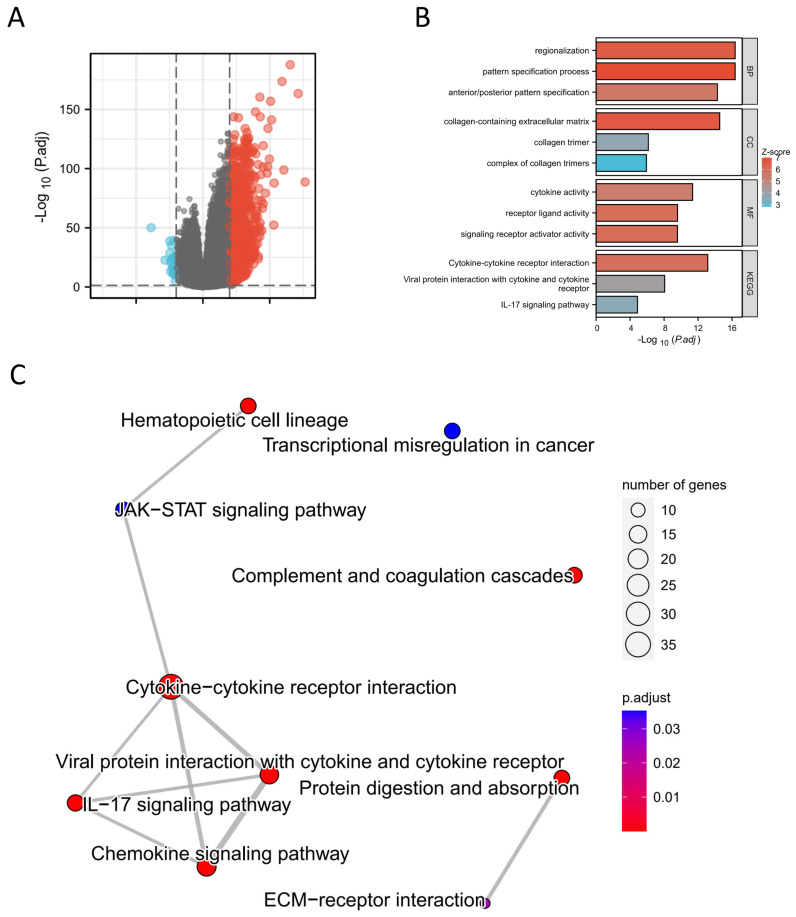
Differential expression analysis of high and low expression TRIM6 groups in Gliomas. (**A**) the volcano map shows the differentially expressed genes (DEGs) between high and low expression TRIM6 groups. (**B**) The barplot demonstrates the KEGG enrichment results of DEGs. (**C**) The cnetplot shows the GO and KEGG analysis of DEGs.

**Figure 6 biomolecules-13-01298-f006:**
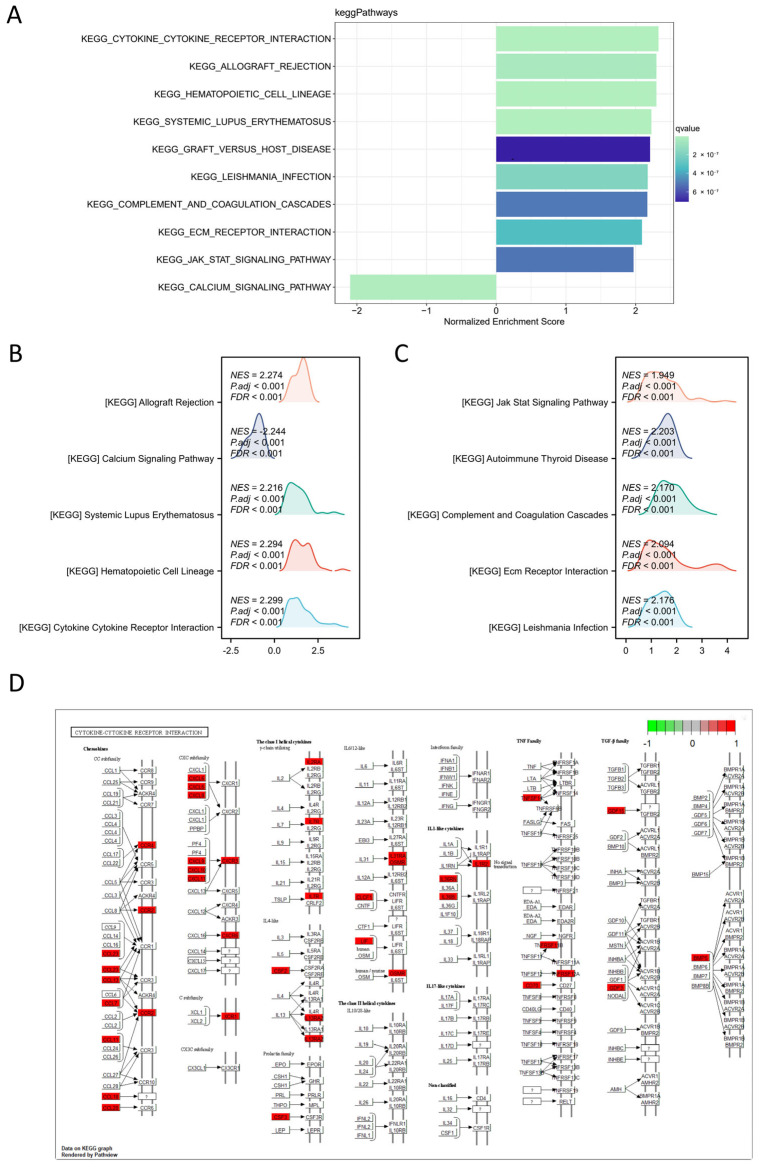
Single gene difference analysis between high expression TRIM6 group and low expression TRIM6 group using data from TCGA glioma database. (**A**) Enrichment plots from gene set enrichment analysis (GSEA). (**B**–**C**) mountain map of top 10 KEGG Gene sets. (**D**) Pathview map of Cytokine-cytokine receptor interaction (map040600).

**Figure 7 biomolecules-13-01298-f007:**
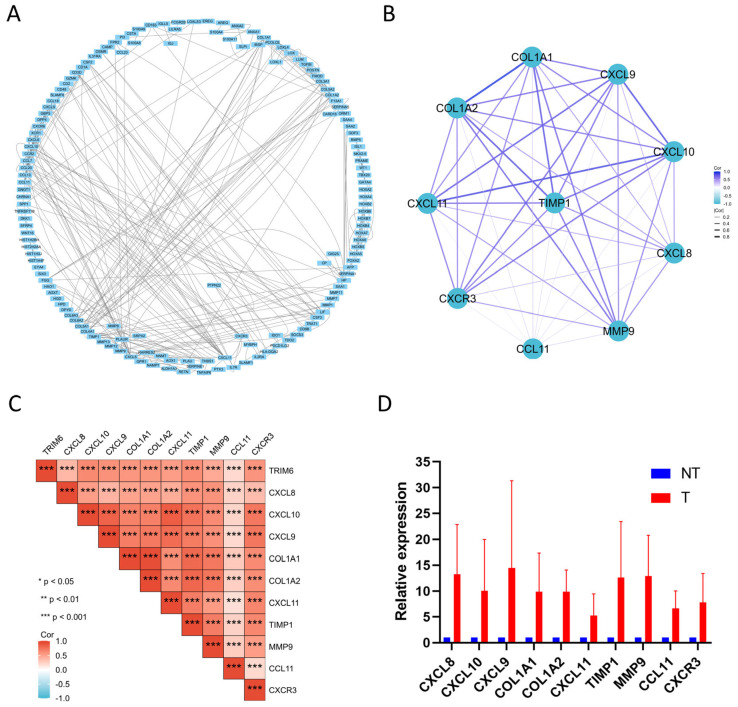
PPI network analysis and correlation analysis between TRIM6 and hub genes using data from TCGA glioma database. (**A**) Visual map of the protein-protein interaction network for high and low expression TRIM6 groups. (**B**) Cytoscape’s plug-in cytoHubba uses the Degree algorithm to select the hub genes from the PPI network. (**C**) Heatmap demonstrates the correlation between TRIM6 and hub genes. (**D**) Detection of hub genes expression levels in Glioma tumor tissues and corresponding adjacent non-tumor tissues using qPCR experiment. Data are presented as mean ± SD. Statistical analysis was performed using *t*−test, and *p* < 0.05 was considered statistically significant (*** *p* < 0.001).

**Figure 8 biomolecules-13-01298-f008:**
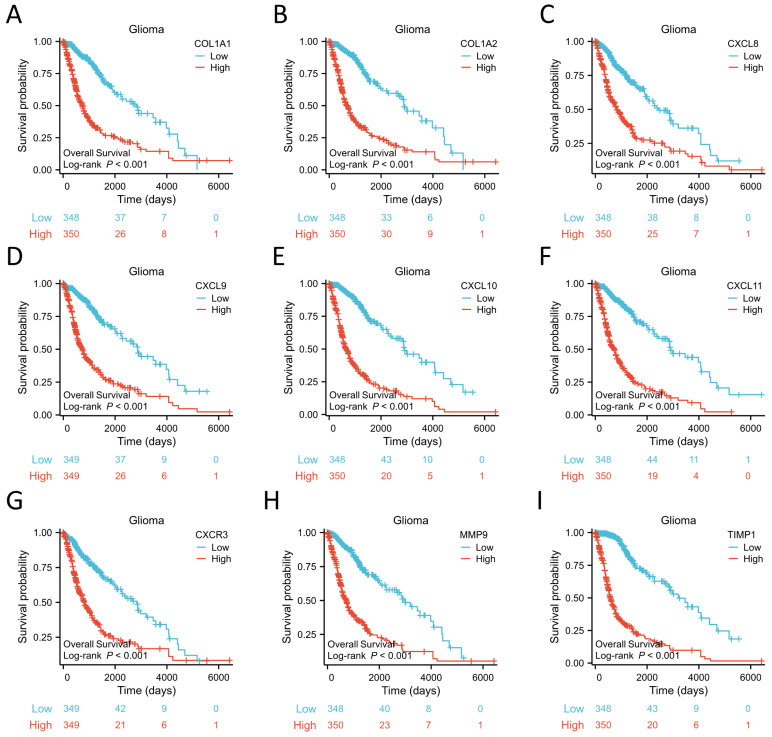
Prognostic analysis of 9 hub genes using data from TCGA glioma database. COL1A1 (**A**), COL1A2 (**B**), CXCL8 (**C**), CXCL9 (**D**), CXCL10 (**E**), CXCL11 (**F**), CXCR3 (**G**), MMP9 (**H**), TIMP1 (**I**).

**Figure 9 biomolecules-13-01298-f009:**
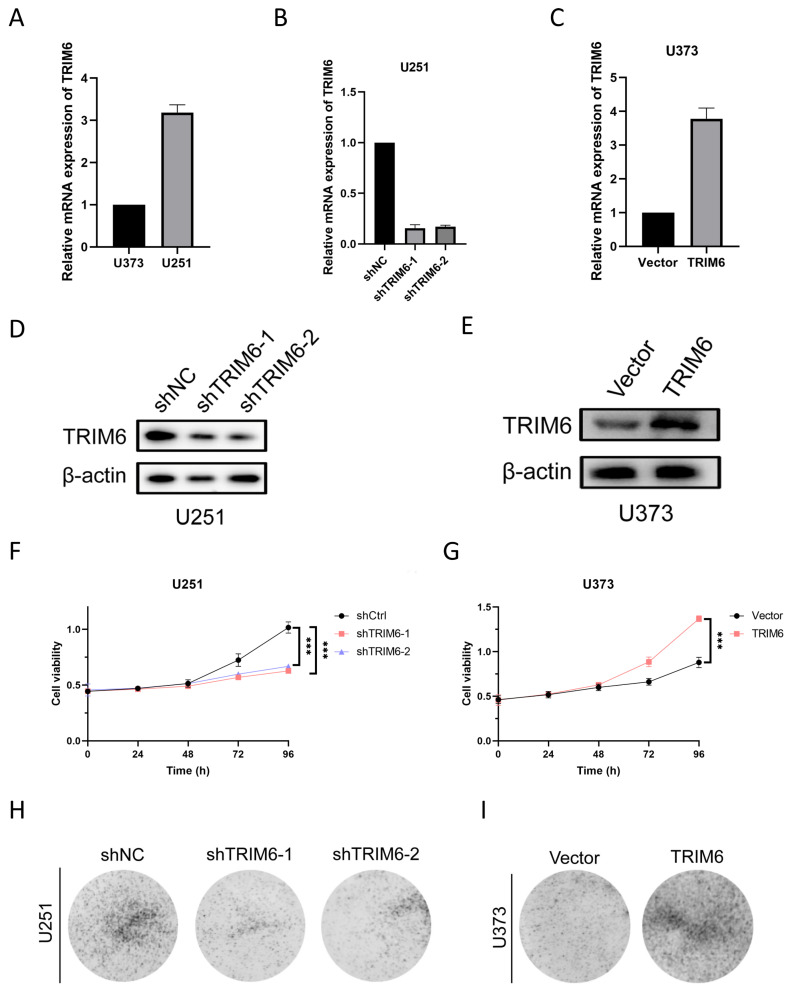
Constructing and validating the effects of TRIM6 knockdown and overexpression on cell proliferation in glioma cell lines. (**A**) Relative mRNA expression levels of TRIM6 in U251 and U373 cells. (**B**,**D**) qPCR and Western blot validation of TRIM6 knockdown effect in U251 cell line. (**C**,**E**) qPCR and Western blot validation of TRIM6 overexpression effect in U373 cell line. (**F**,**G**) The CCK-8 assay was used to detect the cell viability after knocking down TRIM6 in the U251 cell line and overexpressing TRIM6 in the U373 cell line at 24h, 48h, 72h, and 96h. The data represents the mean ± standard deviation (SD) of three independent experiments. (**H**,**I**) The colony formation was evaluated using the plate cloning assay after knocking down TRIM6 in the U251 cell line and overexpressing TRIM6 in the U373 cell line. Data are presented as mean ± SD. Statistical analysis was performed using t-test, and *p* < 0.05 was considered statistically significant (*** *p* < 0.001).

**Figure 10 biomolecules-13-01298-f010:**
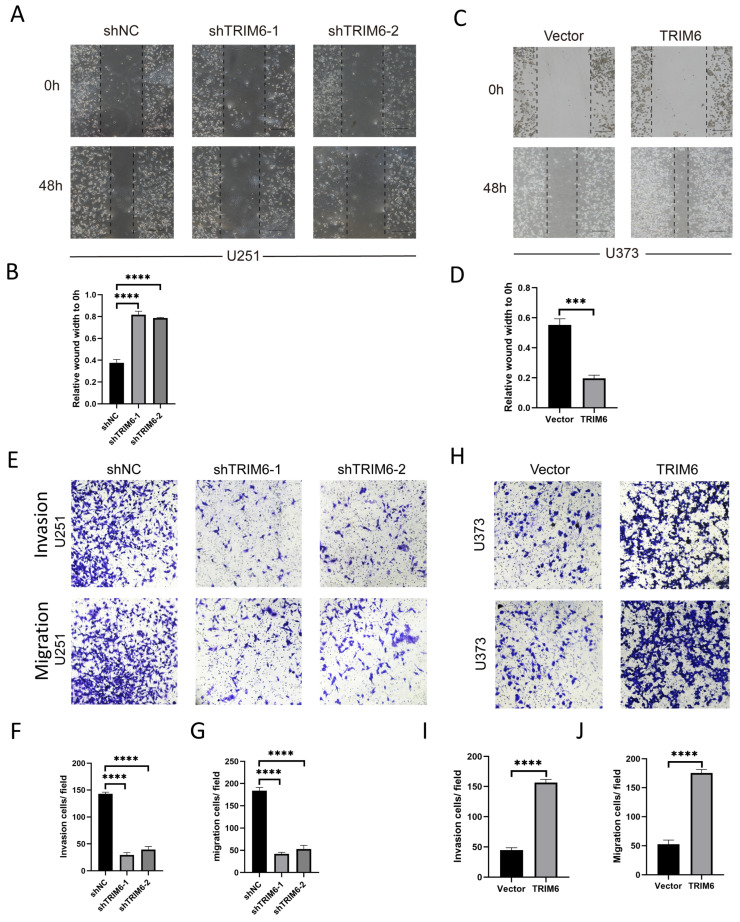
Investigate the effects of knocking down or overexpressing TRIM6 on cell invasion and migration abilities in glioma cell lines. Graphical representation of wound healing assay results is shown in (**A**,**C**). (**B**,**D**) The stacked bar graph demonstrates the relative wound width in U251 cells and U373 cells in various treatment groups, respectively. Graphical representation of the Transwell assay results is shown in (**E**,**H**). The upper panel illustrates the migration of cells across the Transwell membrane, while the lower panel depicts the invasion of cells through the Matrigel-coated membrane. (**F**,**I**) Invasion: The stacked bar graph demonstrates the relative number of invasive U251 cells in various treatment groups. (**G**,**J**) Migration: The bar graph represents the relative number of migrated U373 cells in different experimental conditions. Data are presented as mean ± SD. Statistical analysis was performed using *t*-test, and *p* < 0.05 was considered statistically significant (*** *p* < 0.001, **** *p* < 0.0001).

**Figure 11 biomolecules-13-01298-f011:**
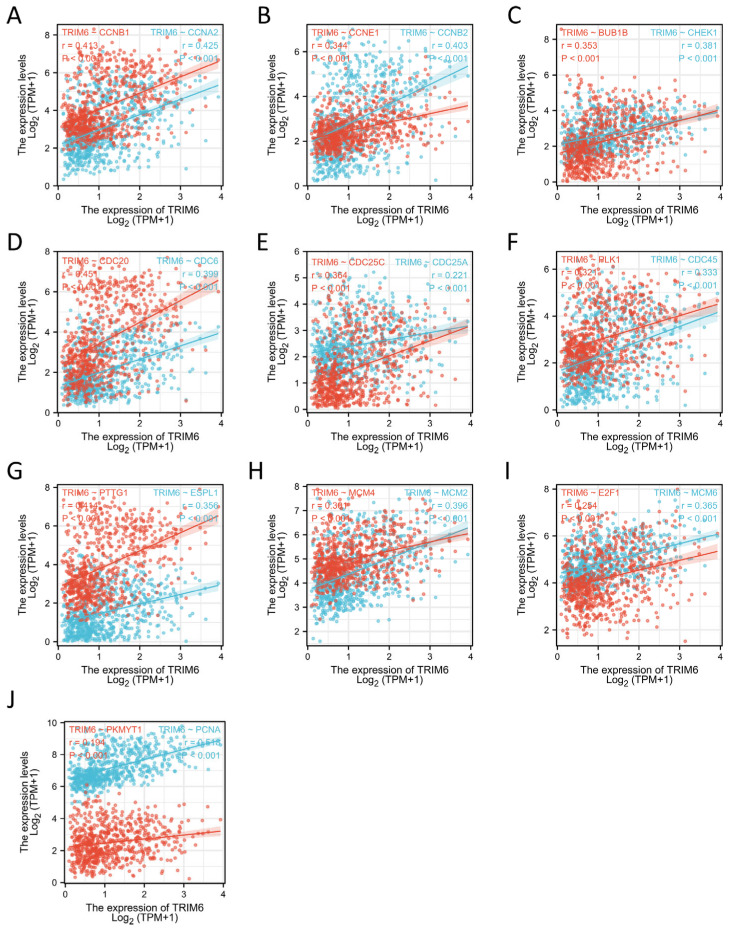
Correlation analysis between TRIM6 and the cell cycle regulatory genes in Gliomas in The Cancer Genome Atlas (TCGA). (**A**) CCNA2, CCNB1, (**B**) CCNB2, CCNE1, (**C**) CHEK1, BUB1B, (**D**) CDC6, CDC20 (**E**) CDC25A, CDC25C, (**F**) CDC45, PLK1, (**G**) EXPL1, PTTG1, (**H**) MCN2, MCM4, (**I**) MCM6, E2F1, (**J**) PCNA, PKMYT1. r, Pearson correlation coefficient.

**Figure 12 biomolecules-13-01298-f012:**
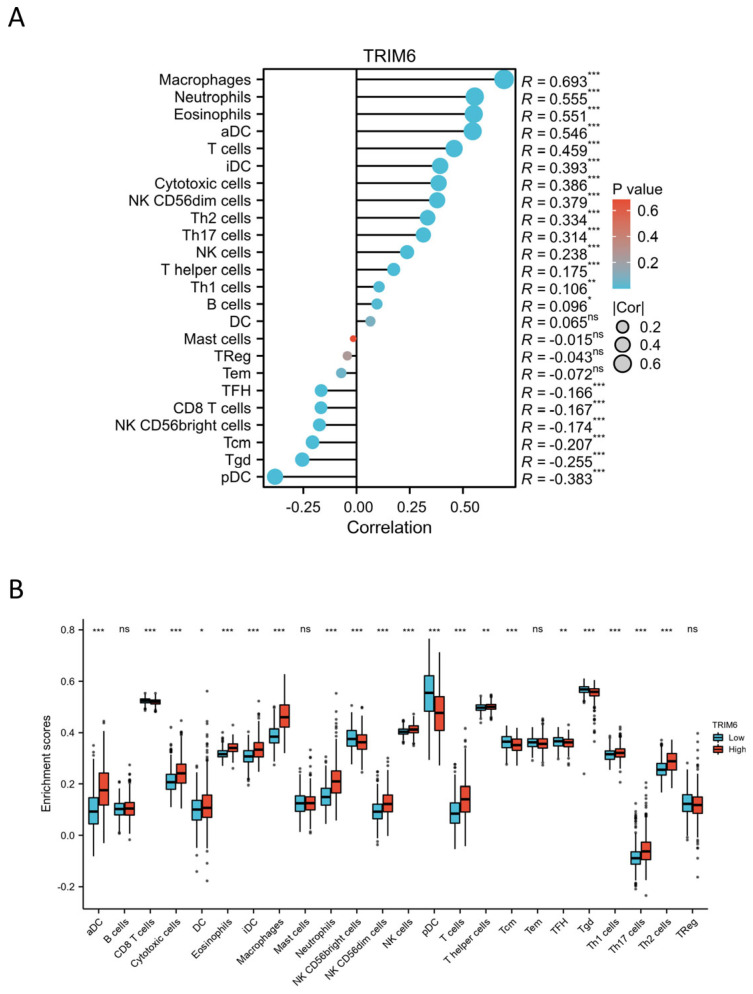
Correlation analysis of TRIM6 expression and immune infiltration in Gliomas. (**A**) Correlation between TRIM6 expression level and Glioma immune cell infiltration level. (**B**) Group comparison between high and low expression TRIM6 groups in Glioma immune cell immersion. Data are presented as mean ± SD. *p* < 0.05 was considered statistically significant (* *p* < 0.05, ** *p* < 0.01, *** *p* < 0.001).

**Table 1 biomolecules-13-01298-t001:** Features of the TCGA Glioma patients based on the TRIM6 expression.

Characteristic	Low Expression of TRIM6	High Expression of TRIM6	*p*
n	348	348	
WHO grade, n (%)			<0.001
G2	153 (24.1%)	71 (11.2%)	
G3	139 (21.9%)	104 (16.4%)	
G4	18 (2.8%)	150 (23.6%)	
IDH status, n (%)			<0.001
WT	31 (4.5%)	215 (31.3%)	
Mut	314 (45.8%)	126 (18.4%)	
1p/19q codeletion, n (%)			<0.001
codel	124 (18%)	47 (6.8%)	
non-codel	223 (32.4%)	295 (42.8%)	
Gender, n (%)			0.026
Female	164 (23.6%)	134 (19.3%)	
Male	184 (26.4%)	214 (30.7%)	
Age, n (%)			<0.001
≤60	312 (44.8%)	241 (34.6%)	
>60	36 (5.2%)	107 (15.4%)	
Histological type, n (%)			<0.001
Astrocytoma	109 (15.7%)	86 (12.4%)	
Glioblastoma	18 (2.6%)	150 (21.6%)	
Oligoastrocytoma	77 (11.1%)	57 (8.2%)	
Oligodendroglioma	144 (20.7%)	55 (7.9%)	
Age, median (IQR)	40 (32.75, 51)	53 (38, 63)	<0.001

**Table 2 biomolecules-13-01298-t002:** Cox regression analyses to explore the independent indicators of OS in Gliomas.

Characteristics	Total(N)	Univariate Analysis		Multivariate Analysis	
		Hazard Ratio (95% CI)	*p* Value	Hazard Ratio (95% CI)	*p* Value
WHO grade	634				
G2	223	Reference			
G3	243	2.999 (2.007–4.480)	**<0.001**	1.770 (1.104–2.837)	**0.018**
G4	168	18.615 (12.460–27.812)	**<0.001**	5.070 (1.567–16.398)	**0.007**
IDH status	685				
WT	246	Reference			
Mut	439	0.117 (0.090–0.152)	**<0.001**	0.498 (0.285–0.870)	**0.014**
1p/19q codeletion	688				
codel	170	Reference			
non-codel	518	4.428 (2.885–6.799)	**<0.001**	1.002 (0.517–1.942)	0.996
Primary therapy outcome	461				
PD	112	Reference			
SD	147	0.440 (0.294–0.658)	**<0.001**	0.361 (0.214–0.608)	**<0.001**
PR	64	0.170 (0.074–0.391)	**<0.001**	0.189 (0.067–0.534)	**0.002**
CR	138	0.133 (0.064–0.278)	**<0.001**	0.167 (0.077–0.364)	**<0.001**
Gender	695				
Female	297	Reference			
Male	398	1.262 (0.988–1.610)	0.062	1.651 (1.049–2.598)	**0.030**
Age	695				
<=60	552	Reference			
>60	143	4.668 (3.598-6.056)	**<0.001**	4.071 (2.438–6.796)	**<0.001**
Histological type	695				
Astrocytoma	195	Reference			
Glioblastoma	168	6.791 (4.932–9.352)	**<0.001**		
Oligoastrocytoma	134	0.657 (0.419–1.031)	0.068	1.132 (0.655–1.956)	0.657
Oligodendroglioma	198	0.580 (0.395–0.853)	**0.006**	0.612 (0.347–1.078)	0.089
TRIM6	695				
Low	347	Reference			
High	348	4.023 (3.077–5.261)	**<0.001**	1.591 (1.027–2.466)	**0.038**

**Characteristics**: Variables and Groupings. **Total (N)**: Number of samples in each variable’s total selected group and its respective subgroups. This represents the overall sample size for each variable and corresponding grouping, used for conducting univariate analysis. **HR (95% CI) Univariate analysis**: Hazard Ratio (HR) values obtained from the univariate analysis along with their corresponding confidence intervals (CIs). The “Reference” category represents the reference group for categorical variables, while other groups are compared to this reference group. ***p* value Univariate analysis**: *p*-value associated with the independent variable obtained from the univariate analysis. If it meets a predetermined threshold, it is considered significant and included in the multivariable model. **HR (95% CI) Multivariate analysis**: Only variables meeting the predefined *p*-value threshold (0.1) for inclusion in the multivariable Cox model will have values reported here. The Glioblastoma was not included in the multivariable regression analysis due to its collinearity with other variables. PD: progressive disease. SD: stable disease. PR: partial response. CR: complete response.

## Data Availability

The original contributions presented in the study are included in the article/Appendix A. Further inquiries can be directed to the corresponding authors.

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
