# Peer review of "TRIM6: An Upregulated Biomarker with Prognostic Significance and Immune Correlations in Gliomas"

_biomolecules, 2023, doi:10.3390/biom13091298_

Round 1

Reviewer 1 Report

The manuscript by Guo Jianrong and Colleagues explores the role of TRIM-6, a member of the tripartite-motif containing protein family, in the development and progression of gliomas. These are tumors of the central nervous system, showing different severity, according to the WHO classification. The article is easy to follow, it is very well written and organized, moreover experiments are thoroughly descripted, so that the logic behind is clear. However, in some instances the reader would benefit form more details, in particular regarding the techniques, so that the experiments could be more replicable. Hence, I have some concerns that the Authors should resolve, before recommending the article for publication. My comments are detailed below.

1.     Line 68, 77 and throughout the manuscript: please define acronyms the first time they are used in the text (also in Supplementary tables)– I guess WNV is West Nile Virus, but this may not obvious to all readers. Please put the list of abbreviation at the end of the manuscript in alphabetical order, so that it is easier to read.

2.     Line 139: please add details about the Ethical committee, for example the approval number/code. Also, at line 141, who approved the guidelines? The institution where experiments were carried out? NIH? The Ministry of Health?...

3.     Line 146: 100 g/ml Penicillin/Streptomycin does not make sense. Line 150: which lentiviral construct was used?

4.     Line 168-169: please add catalogue number for antibodies

5.     In paragraphs 2.11 and 2.12, please check superscript/formatting…: for example, line 173 and 177 and 181 (3x103 cells/well should be 3x10^3… And so on), lines 174, 183 and 193 (CO2 should be CO2). Line 208: ‘ss’ should be ‘is’.

6.     Line 195: which software was used for cell quantification?

7.     Line 204: why using both uni- and multivariate analyses? When is it right to use uni- or multi-variate?

8.     From line 266 to 272: it appears that high TRIM6 is not correlated with worse survival in glioblastoma patients: please specify the number for Grade 4 (now reports only for grades 2 and 3) and for histological diagnosis ‘Glioblastoma’: even if not significant, this could be one of the major results of the paper, since it may differentiate prognosis for the different tumor types. These results should be adequately discussed in the Discussion chapter.

9.     Cell lines experiments (line 143 and 356 onwards): please specify why 2 different cell lines were used, why these 2 lines (U251 and U273) were chosen and at how many passages (from the certified batch, acquired from a certified source) did the cells, before making the experiments. I am quite concerned about this, since U251 and U273 may be synonymous: see for example a common supplier in other countries: https://www.sigmaaldrich.com/IT/it/product/sigma/cb_09063001?gclid=EAIaIQobChMIpvyx4rTHgAMVEa53Ch0UpwsKEAAYAiAAEgIkW_D_BwE&gclsrc=aw.ds Here is the text: 

“Derived from a malignant glioblastoma tumour by explant technique. U-251 was formerly distributed as U-373 MG (Sigma product no. 89081403) until short tandem repeat (STR)-PCR profiling confirmed identity with U-251. A new deposit of U-373 MG known as U-373 MG (Uppsala) is now available (Sigma product no. 08061901). Background to the identity query for the cell line U-373 MG: The American Type Culture Collection (ATCC) reported that their stock of U-373 MG had been shown to have differing genetic properties to stock from the originator′s laboratory, and to share similarities with another glioblastoma cell line, U-251. In light of this, ECACC undertook an investigation into the authenticity of its own stock of the U-373 MG cell line. ECACC found similar results to the ATCC i.e., the stock held as U-373 MG was found to be identical by STR-PCR profiling to U-251. The U-373 MG cell line listed under product no. 89081403 has been re-named as ′U-251 (formerly known as U-373 MG)′ and has the new Sigma product no. 09063001.” Since the authors report in Fig 9 a very different TRIM6 expression in their 2 cell lines, it would be interesting to know exactly the identity of their cells, with reference to commonly accepted standards (e.g. ATCC). I guess the cell provider may provide to the Authors all the details about these two lines.

10.  Figure legends: it would help the readers to have the abbreviations made explicit in the figure legends. Please add also in all the figure legends the meaning of the Box-and-whisker or error bars: do the box represent mean or median +/- 95% confidence intervals? Or Sd, or SEM? Do error bars represent in histograms the SEM or SD? 

11.  In Figure 3, please add the Kaplan-Meyer curves for Grade 4 (after panel I) and GBM (after panel M) and use the same X axis for sub-groups (or the same for all, e.g.: 200 months)

12.  Figure 5 Is legend for B and C reversed? The bar plot is in B, not in C.

13.  Legend to Figs. 6 and 8: specify which tumors/databases.

14.  Figure 7: specify which tumors/databases. Panel A is very nice but cannot be read: please add also a table (maybe in Supplementary material) with the list of all interacting proteins (besides improving the resolution of the image). Legend to panel 7C: what is the ‘correction’ demonstrated? 7D: what are the bars (mean? Medians?) and error bars (SD, SEM…)?

15.  Fig 8 D/E: add the full original Western blot in the Supplementary.

16.  Fig 10 F/I legend: Is really the ‘percentage’ that is represented in the graph? Or the ‘number’ of cells?

17.  Fig 12 A legend: does the bar chart show a Correlation? Where is ‘cell infiltration level’? C and D legends are missing. What’s the meaning of ‘immune cell immersion’ (line 492)?

18.  Why in Table 2 some lines are missing? I mean, multivariate analysis for glioblastoma. What is multivariate analysis comparing for dichotomic variables (e.g., Gender)? The same applies to Supplementary Tables 1 and 2.

19.  Discussion, line 568: the data do not demonstrate that TRIM6 ‘regulate’ the cell cycle. They show that TRIM6 change, but no causal relationship is demonstrated here. The same at line 590, ‘causing immunomodulation’, better to say ‘affecting immunomodulation’

Well written, only very minor issues detected and reported in the main comments. 

Reviewer 2 Report

Minor edits: 

1. Line 569: Tumor microenvironment and surrounding environment: vague and repetitive. What do you mean? Tumor microenvironment constitutes the surrounding environment. 

2. 582-583: generalized statement, maybe misleading. What do you mean by regulatory role of TRIM6 in the development of tumor microenvironment: Likely correlation with certain immune populations? Role of TRIM6 in development of glioma is supported by your data. (that part of the statement is fine)

Round 2

Reviewer 1 Report

The Authors solved any issue I raised, hence the manuscript can be accepted.